# Random Collective Representation-Based Detector with Multiple Features for Hyperspectral Images

**Zhongheng Li [1]**, **Fang He [2]**, **Haojie Hu [2]**, **Fei Wang [1,\*]** and **Weizhong Yu [1]**

[1] School of Electronic and Information Engineering, Xi'an Jiaotong University, Xi'an 710049, China; lizhongheng2010@gmail.com (Z.L.); yuwz05@mail.xjtu.edu.cn (W.Y.)

[2] Xi'an Research Institute of Hi-Tech, Xi'an 710025, China; fanghe1107@gmail.com (F.H.); haojiehu705@gmail.com (H.H.)

[\*] Correspondence: wfx@mail.xjtu.edu.cn

**Abstract:** Collaborative representation-based detector (CRD), as the most representative anomaly detection method, has been widely applied in the field of hyperspectral anomaly detection (HAD). However, the sliding dual window of the original CRD introduces high computational complexity. Moreover, most HAD models only consider a single spectral or spatial feature of the hyperspectral image (HSI), which is unhelpful for improving detection accuracy. To solve these problems, in terms of speed and accuracy, we propose a novel anomaly detection approach, named Random Collective Representation-based Detector with Multiple Feature (RCRDMF). This method includes the following steps. This method first extract the different features include spectral feature, Gabor feature, extended multiattribute profile (EMAP) feature, and extended morphological profile (EMP) feature matrix from the HSI image, which enables us to improve the accuracy of HAD by combining the multiple spectral and spatial features. The ensemble and random collaborative representation detector (ERCRD) method is then applied, which can improve the anomaly detection speed. Finally, an adaptive weight approach is proposed to calculate the weight for each feature. Experimental results on six hyperspectral datasets demonstrate that the proposed approach has the superiority over accuracy and speed.

**Keywords:** hyperspectral image (HSI); hyperspectral anomaly detection (HAD); multiple feature; collaborative representation-based detector (CRD); ensemble and random collaborative representation detector (ERCRD); random collective representation-based detector with multiple feature (RCRDMF)

## 1. Introduction

Hyperspectral imagery (HSI), a term that refers to images captured by hyperspectral sensors, can provide rich spectral information for use in identifying different materials with thousands of adjacent contiguous electromagnetic spectrum bands [1–5]. Due to its fine detection capability, HSI has been widely applied in the fields of target detection [2], change detection [6–8], and classification [9–11]. Of these, target detection plays a particularly important role in many different fields, such as environmental monitoring [12–14] and mineral exploration [15]; accordingly, in recent years, many scholars have begun conducting research in this field [2].

Depending on whether or not prior information is used, target detection methods can be divided into supervised and unsupervised methods [16]. Supervised target detection methods utilize the known spectral information to detect targets [17], while unsupervised methods detect anomalies from their surrounding background without any prior information [2,18]. In fact, it is difficult to acquire the accurate spectral information of ground targets, because it is susceptible to atmosphere absorption, illumination change, noise corruption, and so on [19]. Thus, it is difficult to obtain the precise spectral information for the supervised method. However, unsupervised target detection methods, which are

also referred to as anomaly detection methods and require no prior information, have more important research value in reality and have, thus, been widely applied in various fields, including food quality, safety control, search and rescue, mineral detection, and environmental surveillance [2,20–22].

Anomaly detection also can be viewed as a binary classification problem, in which it is necessary to separate the anomaly class from the background class [1,2]. The anomaly pixels typically exhibit distinct spectral or spatial differences with their surrounding background [2,23–25]. A number of anomaly detection methods have been developed in recent years, among the most famous of which is the Reed-Xiaoli (RX) method. First, the RX method assumes that the background conforms to a multivariate Gaussian distribution, after which the corresponding mean and covariance values of the HSI image can be calculated. Finally, an appropriate threshold is set to distinguish the anomalies by estimating the Mahalanobis distance [26]. The global RX (GRX) and local RX (LRX) are two versions of the RX algorithm. The GRX method uses the whole image to model the background, while LRX utilizes the local dual-window to model the background [27]. The kernel RX (KRX) [28] algorithm is a nonlinear version of the RX algorithm that uses kernel theory to transform every pixel to a high-dimensional space. However, KRX is highly computationally demanding, making it unsuitable for the processing of large-scale hyperspectral data. A modified KRX method has also been proposed to improve the calculation efficiency of KRX algorithm, which assumes that the background class is a spherical covariance matrix [29]. Moreover, a fast generalization of KRX, namely cluster KRX (CKRX), was also proposed. This approach applies a fast eigendecomposition method to achieve anomaly detection by clustering the background pixels [30].

In addition, a number of representation-based methods have been proposed. These methods do not require any statistical assumptions and have, thus, attracted significant attention [20]. These techniques make use of the conspicuous characteristics of anomalies: the low probability of occurrence and the different spectral signature from the background pixels [27]. There are several forms of representation-based methods, including the sparse representation-based methods [1,31–37], the low-rank methods [38–43], and the collaborative methods [44–48].

Recently, sparse representation methods have been widely used in hyperspectral anomaly detection applications. Li et al. proposed using background joint sparse representation (BJSR) model for hyperspectral anomaly . First, the BJSR model is adopted to estimate the adaptive orthogonal background complementary subspace through adaptively selective the most representative background bases for the local region. An unsupervised adaptive subspace detection method is then proposed to control the influence of the background while highlighting the anomalies [34]. Ma et al. designed a new spectral feature selection framework based on sparse representation for anomaly detection. The residues between the background spectrum reconstruction error and anomaly spectrum recovery error are minimized so as to enable the representative spectra to be picked out. In this way, the anomaly's deviation can be significantly enlarged relative to the background. Finally, a global multiple-view detection strategy is presented that can improve the detection accuracy by comprehensively considering the virtues of different groups of representative features selected from multiple dictionaries [36]. Ling et al. proposed a hyperspectral anomaly detection method that operates by sparse representation and linear mixture model (SR-LMM). This algorithm assumes that the background can be approximately represented as a sparse linear combination of its surroundings, while the anomaly cannot [1].

Moreover, a number of studies have also investigated low-rank representation-based anomaly detection approaches, which assume that the background follows a low-rank prior and the anomaly is sparsely distributed [38]. Sun et al. proposed the low-rank and sparse matrix decomposition (LRaSMD) approach, which calculates the Euclidean distance between the corresponding sparse component vector and the mean vector of the sparse matrix, an approach enabling the score of each pixel to be obtained [40]. Based on the LRaSMD model, some other improved algorithms have been proposed. Zhang et al. proposed an

LRaSMD-based Mahalanobis distance method (LSMAD). This approach utilizes the low rank of the background together with the sparse property of the anomalies, enabling the background and the sparse component to be obtained. The low-rank prior knowledge of the background is explored to facilitate computing of the background statistics and constructs a Mahalanobis-distance-based anomaly detector [41]. Zhang et al. proposed the parts representation-based low rank and sparse matrix decomposition anomaly detector (PRLRaSAD), which assumes that the low rank component can be described via parts-based representation. Furthermore, PRLRaSAD combines parts-based and holistic-based representation to model the original HSI. Owing to the sparse properties of the anomaly target, it is grounded in a holistic-based representation, while the background is grounded in parts-based representation. Based on these descriptions of HSI, the PRLRaSAD method divides the HSI decomposition optimization problem into three subproblems, so that the basis vector matrix, coefficient matrix, and sparse matrix, respectively, can be computed [42].

Window-based operations are the most commonly used technique for hyperspectral anomaly detection. Among them, collaborative representation-based detector (CRD) is the most representative anomaly detection method, and it was first used for hyperspectral anomaly detection in Reference [44]. The CRD algorithm assumes that each pixel in the background can be approximately represented by its neighborhoods, while this is not the case for anomalies. A sliding dual window is used to achieve background estimation. For the CRD method, when more classes (i.e., samples for anomaly detection) are involved, the least squares solution is unstable. Moreover, if the test pixel is anomalous, while several samples from the background are similarly anomalous, a detection error will occur. To solve the limitations of CRD algorithms, Su et al. proposed the CRD with principal component analysis (PCA) for removing outlier (PCAroCRD) model, which adopts spatial-domain PCA to exact the principal information of background. This principal information was then used as samples for collaborative representation, while the information of abnormal pixels in the background was removed. Adopting this approach can make the detection result more stable [45]. Vafadar et al. proposed a modified collaborative-representation-based method with outlier removal anomaly detector (CRBORAD), which utilizes spectral information together with spatial information to detect anomalies. The CRBORAD method adaptively estimates the background with reference to its adjacent pixels within a sliding dual-window. Through subsequent stages, precise anomaly detection can be obtained [46]. Zhang et al. proposed a dual collaborative representation (DCR)-based hyperspectral anomaly detection method to resolve a common problem, namely that the attributes of test pixels are often affected by the background knowledge containing abnormal information. The DCR method employs low-rank and sparse matrix decomposition to obtain a low-rank background matrix. The density information of the pixels in a sliding dual window is then calculated by applying the density peak clustering algorithm to the low-rank background matrix. With reference to the pixel density, the highest density can be selected as the pure background pixel set to provide an approximate representation of the test pixels. Based on the residuals of this dual-stage collaborative representation, a decision function can be utilized to detect abnormal pixels [48]. However, these CRD methods adopt the sliding dual-window approach, which introduces higher computational complexity. To solve this problem, Wang et al. proposed the ensemble and random collaborative representation detector (ERCRD), which adopts a random background modeling to replace the sliding dual window of the original CRD [49]. However, the ERCRD algorithm only uses the spectral characteristics of HSI. In reality, HSI images contain rich information: specifically, spectral, textural and spatial features. Making full use of these features can effectively improve the anomaly detection results. The Gabor feature can typically be used to represent the HSI spatial texture information. The extended morphological profile (EMP) and extended multiattribute profile (EMAP) features can further be utilized to represent the HSI spatial structure information.

In this paper, we propose a novel anomaly detection approach, named Random Collective Representation-Based Detector with Multiple Feature (RCRDMF). First, the

four different features, named spectral feature, Gabor feature, EMP feature, and EMAP feature matrix, are extracted from the HSI image. This prevents any single feature from containing only a specific spectral or spatial characteristic of the HSI image. The ERCRD method is then applied to rapidly complete anomaly detection for each feature matrix. In order to simultaneously use the above information from various feature matrix, we need to find an approach to integrate them. A reasonable way is to linearly combine them with appropriate weights. We propose an adaptive weight approach to balance the various feature information, which helps making better use of them for hyperspectral anomaly detection. Compared to the existing CRD-based models, RCRDMF algorithm has the following advantages:

1   The different features contain the spatial characteristics and specific spectral of the HSI. Fusing these features into the anomaly detection model is beneficial to improving the detection accuracy.
2   The ERCRD algorithm can accelerate the speed of anomaly detection. With the help of the ERCRD method, the RCRDMF model also incurs a lower time cost than the traditional CRD approaches.
3   The adaptive weight approach is proposed to calculate the weight for each feature, which avoids the need to tune the weight parameter.

The remainder of this paper is organized as follows. Section 2 briefly reviews the work of CRD and ERCRD and presents the details of the proposed RCRDMF. Experimental results are presented in Section 3 and discussed in Section 4. Finally, our conclusions are presented in Section 5.

## 2. Materials and Methods

Let $X \in \mathbb{R}^{d \times n}$ denote the HSI data, where $d$ and $n$ represent the number of dimensions and pixels, respectively.

### 2.1. Collective Representation-Based Detector

The CRD assumes that the background point can be linearly represented by adjacent pixels, while an anomaly cannot. Let $X = [x_1, x_2, \cdots, x_n] \in \mathbb{R}^{d \times n}$ represent a two-dimensional HSI matrix that is transformed via three-dimensional hyperspectral imagery, where $n$ and $d$ represent the number of the pixels and spectral bands, respectively. For the pixel $x_i \in \mathbb{R}^{d \times 1}$, its surrounding pixels are selected as adjacent pixels by two square windows of different sizes around the pixel of interest, as shown in Figure 1. The adjacent pixels between the sliding dual window can be represented as $X_s = [\tilde{x}_1, \tilde{x}_2, \cdots, \tilde{x}_s] \in \mathbb{R}^{d \times s}$, where $s$ is the number of the adjacent pixels and $s = w_{out} \times w_{out} - w_{in} \times w_{in}$. The objective function of CRD can, thus, be defined as follows:

$$\min_{\boldsymbol{\alpha}_i} \|x_i - X_s \boldsymbol{\alpha}_i\|_2^2 + \lambda \|\boldsymbol{\alpha}_i\|_2^2, \tag{1}$$

where $\boldsymbol{\alpha}_i \in \mathbb{R}^{s \times 1}$ and $\lambda$ represent the coefficient vector and regularization parameter, respectively. Moreover, by setting the derivative of $\boldsymbol{\alpha}_i$ to zero, the solution of Eq. (1) can be expressed as follows:

$$\hat{\boldsymbol{\alpha}}_i = (X_s^T X_s + \lambda I)^{-1} X_s^T x_i, \tag{2}$$

where $I$ is the identity matrix.

Once the coefficient vector $\boldsymbol{\alpha}_i$ is obtained, the corresponding anomaly score can be computed as follows:

$$\delta_i = \|x_i - X_s \hat{\boldsymbol{\alpha}}_i\|_2. \tag{3}$$

If $\delta_i$ is larger than a predefined threshold, then $x_i$ is regarded as an anomalous pixel.

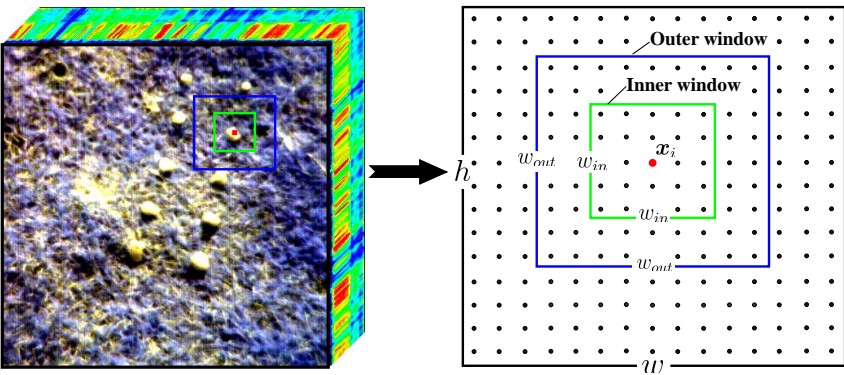

**Figure 1.** The sliding dual window of the collaborative representation-based detector (CRD).

*2.2. Random Collective Representation-Based Detector*

CRD algorithm calculates the optimal weight of each pixel based on the sliding dual window, which incurs very high computational costs. To solve this problem, Wang et al. proposed the Ensemble and Random Collective Representation-Based Detector (ERCRD) method. Rather than using the sliding dual window, some pixels are randomly selected from the whole image as the background. The background pixels can be represented as $X_r = [\tilde{x}_1, \tilde{x}_2, \cdots, \tilde{x}_r] \in \mathbb{R}^{d \times r}$. ERCRD assumes that all non-abnormal points in the image can be linearly represented by these randomly selected background points, an approach that reduces computational complexity. Thus, the objective function is:

$$\min_A \|X - X_r A\|_F^2 + \lambda \|A\|_F^2, \tag{4}$$

where $A \in \mathbb{R}^{m \times n}$ and $\lambda$ are the weight matrix and regularization parameter, respectively. Setting the derivative w.r.t $A$ to zero, the solution of $A$ is:

$$A = (X_r^T X_r + \lambda I)^{-1} X_r^T X. \tag{5}$$

Thus, the matrix $X$ can be reconstructed as $X_r A$. In the same way as CRD, the reconstruction error of pixel $x_i$ can be regarded as the anomaly score, which is obtained in the following way:

$$\delta_i = \|x_i - X_r a_i\|_2, \tag{6}$$

where $x_i$ and $a_i$ are the $i$ column of $X$ and $A$, respectively. Then, if $\delta_i$ is larger than a given threshold, $x_i$ is can be viewed as an anomaly. Figure 2 shows the random background modeling of the ERCRD.

Compared with the CRD method, ERCRD has the following two advantages: faster speed and higher accuracy. However, the ERCRD algorithm uses only the spectral characteristics of HSI, while HSI contains rich information, specifically the spectral feature, texture feature, morphological feature and spatial feature. Making full use of these features can effectively improve the anomaly detection results.

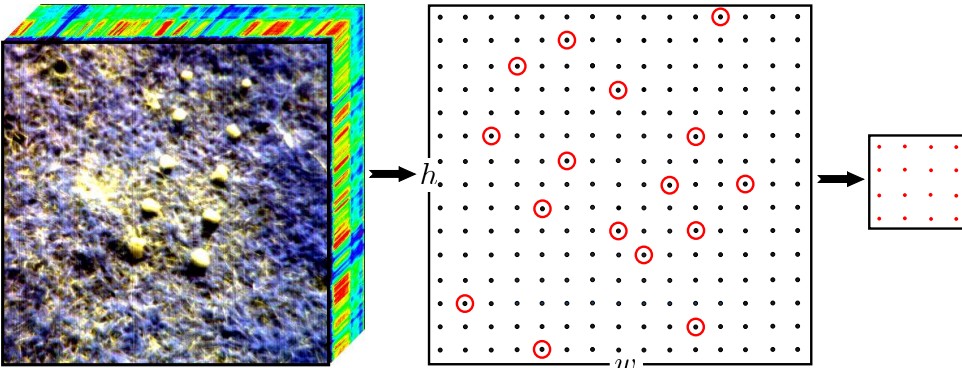

**Figure 2.** Random background modeling of the ensemble and random collaborative representation detector (ERCRD).

*2.3. The Proposed Method*

2.3.1. Multiple Feature Extraction

Usually, there are four types of features that can be extracted from an HSI image: the spectral feature, the Gabor feature, the extended morphological profile (EMP) feature, and the extended multiattribute profile (EMAP) feature. The details of all these features are described as follows:

1. Spectral feature: The spectral feature of pixel $x_i$ is the corresponding spectral signature. We use $X_{spe} \in \mathbb{R}^{d \times n}$ to represent the entire image spectral feature matrix.

2. Gabor feature: The Gabor feature is obtained via Gabor transformation, a transformation method that is capable of extracting the corresponding features from the frequency domain and has been widely used in the image processing field for the extraction of texture features.

    Firstly, a projection matrix $W \in \mathbb{R}^{d \times m}$ should be calculated via the principal component analysis (PCA) model, as follows:

$$\max_{W^T W = I} tr(W^T X_{spe} X_{spe}^T W^T), \tag{7}$$

    where $I \in \mathbb{R}^{m \times m}$ is the identity matrix. $m$ is the number of top principal component images, which can be defined as follows:

$$I_{pci} \in \mathbb{R}^{m \times h \times w} = W^T X. \tag{8}$$

    $I_{pci}$ can then be convolved with a Gabor filter in different orientations and at different scales. The Gabor feature $X_{gabor} \in \mathbb{R}^{d_2 \times n}$ can be then obtained by extracting the filtering coefficients; here, $d_2$ is the product of the number of orientations, scales, and principal component images. For example, when the numbers of orientations, scales, and principal component images are 6, 5, and 5, respectively, $d_2 = 6 \times 5 \times 5 = 150$.

3. EMP feature: The EMP feature is also obtained by means of the PCA (principal component analysis, PCA) method. First, we extract the first $m$ principal component images by means of the PCA approach. Then, the morphological profile of each principal component image is extracted via its structural elements (SEs). Finally, we construct the EMP feature with combining the acquired morphological profiles. For the EMP feature matrix $X_{emp} \in \mathbb{R}^{d_3 \times n}$, $d_3$ is connected to the number of $m$ and SEs. When $m = 5$ and $SEs = 6$, $d_3 = (2 \times 6 + 1) \times 5 = 65$.

4. EMAP feature: The EMAP feature is also based on the top $m$ principal component images. Moreover, it also relies on the morphological attribute filters. For each principal component image, the morphological attribute filters are utilized to generate the extended attribute profiles (EAPs). $EAP = [AP_1, AP_2, \cdots, AP_m]$, where

$AP_i(i = 1, 2, \cdots, m)$ represents the attribute filtering of component $i$. The EMAP feature can be generated by extending EAPs with four different attributes of the regions: area, size, elongation, and homogeneity. In the EMAP feature matrix $\boldsymbol{X}_{emap} \in \mathbb{R}^{d_4 \times n}$, $d_4$ is related to the number of $m$ and the parameters of the attribute filters employed. Since each of filters can produce nine features, $d_4$ can be obtained accordingly, e.g., when $m = 5$, $d_4 = 9 \times 4 \times 5 = 180$.

### 2.3.2. Random Collective Representation-Based Detector with Multiple Feature

Inspired by Ensemble and Random Collective Representation-based Detector (ERCRD) method, we propose a new anomaly detection approach, referred to as Random Collective Representation-Based Detector with Multiple Feature (RCRDMF). First, the four feature matrices discussed above are extracted from the HSI image. Then, for each feature matrix, the ERCRD method is utilized to achieve anomaly detection. Figure 3 shows the feature extract and random background modeling of the RCRDMF. Finally, the adaptive weight approach is adopted to assign the corresponding weight for each feature.

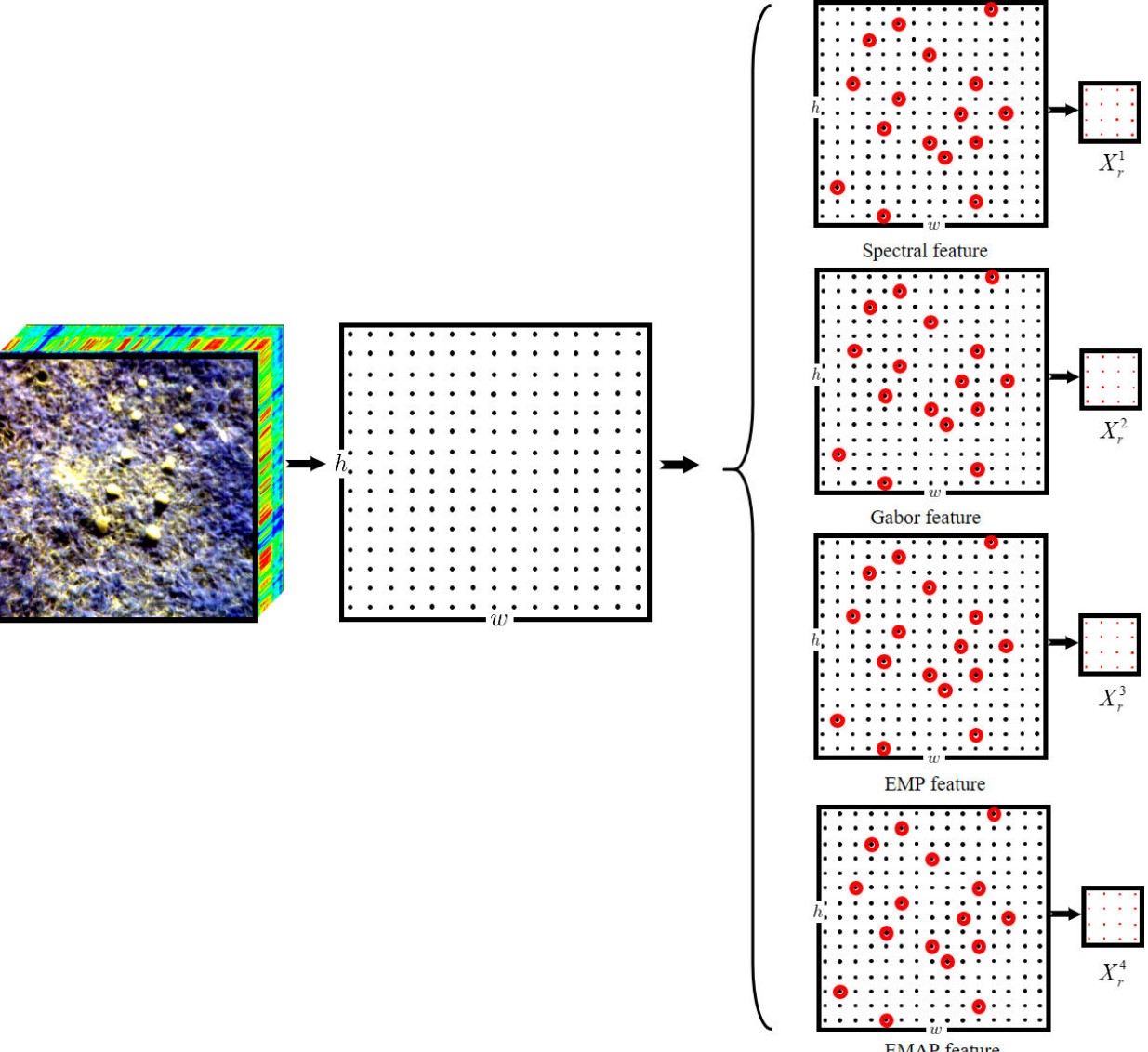

**Figure 3.** Feature extracting and random background modeling of the Random Collective Representation-Based Detector with Multiple Feature (RCRDMF).

Let $X^1 = X_{spe}$, $X^2 = X_{gabor}$, $X^3 = X_{emp}$, $X^4 = X_{emap}$. $X_r^v$ represents the samples in the $v$-th view features, which are randomly selected. The proposed RCRDMF model is then expressed in the following form:

$$\min_{A,\alpha_v} \sum_{v=1}^{V} \frac{1}{\alpha_v} \|X^v - X_r^v A\|_F^2 + \lambda \|A\|_F^2, \quad s.t. \sum_{v=1}^{V} \alpha_v = 1, \alpha_v > 0. \tag{9}$$

$X_r^v \in \mathbb{R}^{d \times r}$ is the randomly selected background in the $v$-th view features, where $r$ represents the number of random sampling pixels. Moreover, $\alpha_v$ is the weight of the $v$-th view features, which can be adaptively solved as the optimization process. $A \in \mathbb{R}^{r \times n}$ and $\lambda$ represent the weight matrix and regularization parameter, respectively.

In Equation (9), there are two variables that must be solved, $A$ and $\alpha_v$. In this paper, we adopt the following alternating iterative approach to solving this problem.

Fix $\alpha_v$, update $A$ Equation (9) can be rewritten in the following form:

$$\min_{A} \sum_{v=1}^{V} \frac{1}{\alpha_v} \|X^v - X_r^v A\|_F^2 + \lambda \|A\|_F^2. \tag{10}$$

Taking the derivative w.r.t $A$, and setting the derivative to zero, we have

$$A = (\sum_{v=1}^{V} \frac{1}{\alpha_v} X_r^{vT} X_r^v + \lambda I)^{-1} (\sum_{v=1}^{V} \frac{1}{\alpha_v} X_r^{vT} X^v). \tag{11}$$

Fix $A$, update $\alpha_v$
Let $h_v = \|X^v - X_r^v A\|_F^2$, such that Equation (9) is equal to:

$$\min_{\alpha_v} \sum_{v=1}^{V} \frac{1}{\alpha_v} h_v, \quad s.t. \sum_{v=1}^{V} \alpha_v = 1, \alpha_v > 0. \tag{12}$$

Thus, the Lagrange function of Equation (12) is:

$$\sum_{v=1}^{V} \frac{1}{\alpha_v} h_v - \lambda_\alpha (\sum_{v=1}^{V} \alpha_v - 1), \tag{13}$$

where $\lambda_\alpha$ is the Lagrange multiplier. With simple algebraic manipulations, we obtain

$$\alpha_v = \frac{(h_v)^{\frac{1}{2}}}{\sum_{v=1}^{V} (h_v)^{\frac{1}{2}}}. \tag{14}$$

Once $A$ and $\alpha_v$ are obtained, the reconstruction error for each pixel $x_i$, which is regarded as the anomaly score, can be calculated as follows:

$$\delta_i = \sum_{v=1}^{V} \frac{1}{\alpha_v} \|x_i^v - X_r^v a_i\|_2. \tag{15}$$

If $\delta_i$ is larger than a given threshold, $x_i$ is can be viewed as an anomaly.

As the background points are randomly selected, the results may be inconsistent at each run. To solve this problem, we adopt the strategy of repeating this process and integrating these results. Define the number of repetitions as $T$; the final anomaly score is as follows:

$$\gamma_i = \sum_{t=1}^{T} \delta_i^t. \tag{16}$$

The steps are presented in more detail in Algorithm 1.

---

**Algorithm 1** The algorithm of RCRDMF

---

**Input:** HSI data matrix $X$, the number of randomly selected background points $r$ and repetitions $T$.

1: Extract the four features: the spectral feature $X^1 = X_{spe}$, the Gabor feature $X^2 = X_{gabor}$, the EMP (extended morphological profile, EMP) feature $X^3 = X_{emp}$, and the EMAP (extended multiattribute profile, EMAP) feature $X^4 = X_{emap}$. Let $X_r^v$ be the samples in the $v$-th view features.
2: **for** $t = 1 : T$ **do**
3:     Initialize weight $\alpha_v = 1/4$.
4:     **repeat**
5:         Fix $\alpha_v$, calculating $A$ by Equation (11).
6:         Fix $A$, calculating $\alpha_v$ by Equation (14).
7:     **until** Converge
8:     Calculate $\delta_i$ by Equation (15)
9: **end for**
**Output:** The final anomaly score by Equation (16).

---

## 3. Experimental Results

In this section, experiments are conducted to verify the detection performance of the proposed RCRDMF approach. All experiments are carried out on a PC with ThinkPad X1 carbon, i7, 16G RAM, MATLAB R2018b. In these experiments, the following six hyperspectral images are chosen.

1.  AVIRIS-I: This image was captured from the San Diego airport area, CA, USA, by the Airborne Visible/Infrared Imaging Spectrometer (AVIRIS) sensor. The original image contained 224 spectral bands in wavelengths ranging from 370 to 2410 nm with 3.5 m pixels. After removing the poor-quality bands, there are 189 bands remaining to be analyzed in the experiments. Moreover, each band has $400 \times 400$ pixels. The AVIRIS-I dataset is selected from the top left of this entire image with a size of $120 \times 120$ pixels. There are three airplanes, containing 58 pixels, which can be regarded as anomalies that should be recognized. The false color image and the corresponding ground truth map of the AVIRIS-I image are shown in Figure 4a,d, respectively.
2.  AVIRIS-II: This image was also obtained from the original San Diego airport image. Unlike the AVIRIS-I dataset, the region selected here is of size $100 \times 100$ pixels and is drawn from the San Diego airport image. In the AVIRIS-II dataset, the anomalies that need to be detected in the scene are also three airplanes, this time taking up a total of 134 pixels. The false color image and the corresponding ground truth map of the AVIRIS-II image are presented in Figure 4b,e, respectively.
3.  AVIRIS-III: As with the AVIRIS-I and AVIRIS-II datasets, it is also selected from the San Diego airport image: the difference is that it is cropped from the top left of the San Diego airport image with a size of $200 \times 240$ pixels. In the AVIRIS-III dataset, six airplanes made up of 90 pixels in total are viewed as the anomalies. The false color image and the corresponding ground truth map of the AVIRIS-III image are shown in Figure 4c,f, respectively.
4.  Cri: It was collected by the Nuance Cri hyperspectral sensor. Cri has 46 spectral bands in wavelengths ranging from 650 to 1100 nm. In each band, there are $400 \times 400$ pixels. The anomalies in this image are 10 rocks containing 2216 pixels. The false color image and the corresponding ground truth map of the Cri image are shown in Figure 4g,j, respectively.
5.  ABU-airport-2: This was acquired by the AVIRIS sensor from ABU(Airport–Beach–Urban) dataset [50]. ABU-airport-2 has 204 spectral bands, each with $100 \times 100$ pixels. In this image, the anomalies are 2 airports. The false color image and the corresponding ground truth map of the ABU-airport-2 image are presented in Figure 4h,k, respectively.

6. Salinas: It was also obtained by the AVIRIS sensor. After discarding the 20 water absorption bands, there are 204 spectral bands remaining in the analysis. The size of each band is $180 \times 180$ pixels. The false color image and the corresponding ground truth map of the Salinas image are shown in Figure 4i,l, respectively.

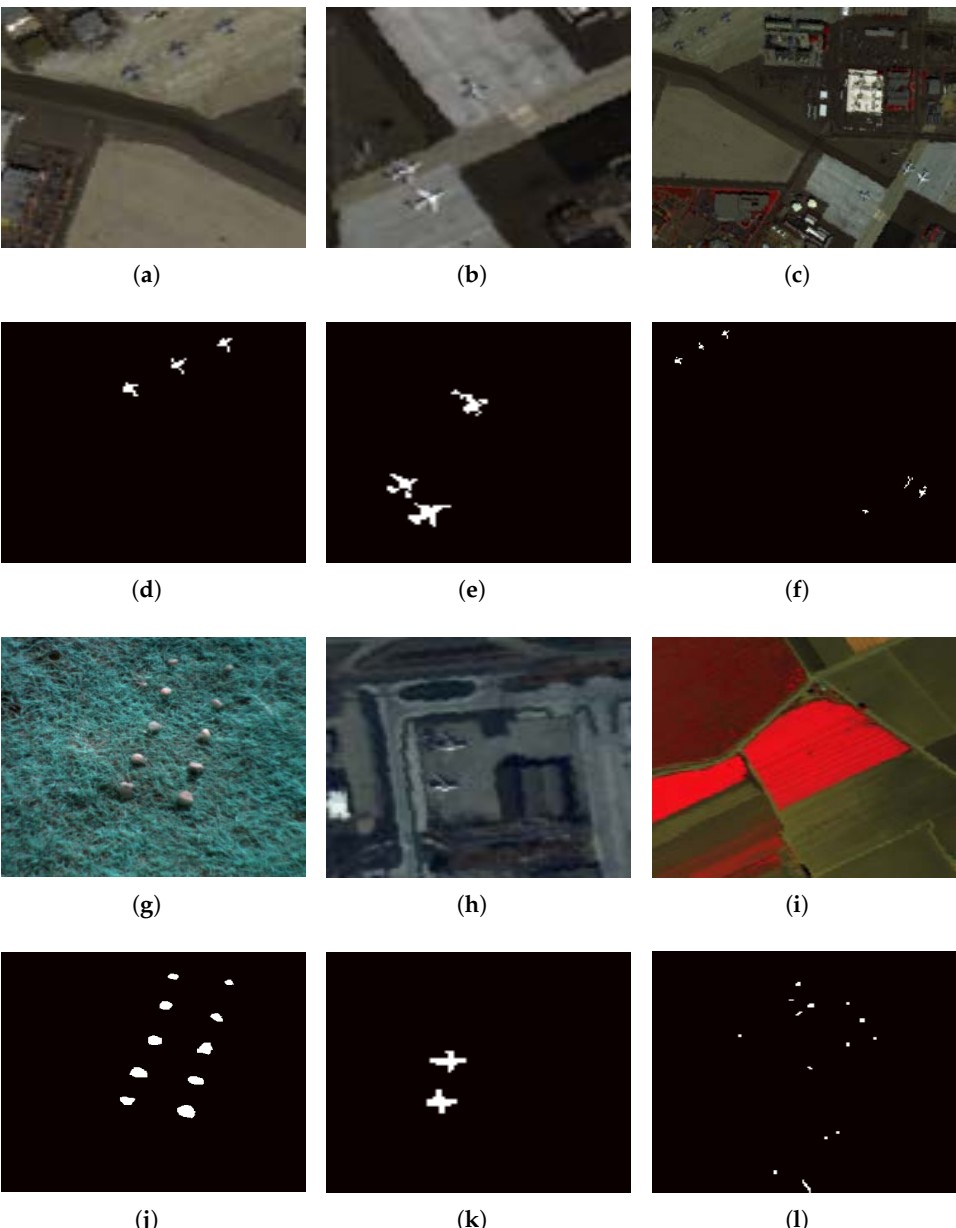

**Figure 4.** False color image of (**a**) Airborne Visible/Infrared Imaging Spectrometer (AVIRIS)-I, (**b**) AVIRIS-II, (**c**) AVIRIS-III. The ground truth map of (**d**)AVIRIS-I, (**e**) AVIRIS-II, (**f**) AVIRIS-III. False color image of (**g**) Cri, (**h**) ABU-airport-2, (**i**) Salinas. The ground truth map of (**j**) Cri, (**k**) ABU-airport-2, (**l**) Salinas.

### 3.1. Detection Performance

In this part, we conduct experiments on the six datasets presented above to evaluate the superiority of the proposed RCRDMF method. The evaluation indicators comprise a qualitative evaluation metric (color detection map) and quantitative evaluation metric (area under the receiver operating characteristic (ROC) curve (AUC) value, running time, ROC curve, and the normalized background-anomaly separation map). The color detection map is a visual representation which uses the brightness to indicate the anomaly detection

result of the pixels. The higher the anomaly detection score, the brighter the pixel, and the more likely the pixel is an anomaly. ROC represents the relationship between the detection probability (DP) and the false alarm rate (FAR). The greater the AUC, the better the detection results. Moreover, the normalized background-anomaly separation maps use two box plots to describe the distribution of anomaly scores $gamma_i$ for anomalies and background points, respectively. The five values in a box plot from top to bottom are the maximum, the upper quartile, the median, the lower quartile, and the minimum. The greater the difference between the distributions of anomalies and background points, the more easily they can be distinguished, thus illustrating the better performance of the algorithm.

We further compare our methods with five state-of-the-art methods: GRX, LRX, CRD, LSMAD, and ERCRD. Among them, LRX and CRD are sensitive to the sliding double window sizes. We set the inner window size $w_{in}$ to range from 3 to 11, while the outer window size $w_{out}$ ranges from 5 to 15. For the ERCRD and RCRDMF methods, the number of random sample $r$ is set based with reference to the dataset size. For small datasets (AVIRIS-I and AVIRIS-II), $r = 10$; for medium datasets (AVIRIS-III, Salinas), $r = 50$; for large datasets (Cri), $r = 100$. Specially, for dataset ABU-airport-2, which is a small dataset, $r = 100$. We will discuss this issue in the discussion section. The number of repetitions is $T = 20$. Table 1 lists the AUC values obtained by different methods, in which the maximum value is in bold. And Table 2 records the detection time.

**Table 1.** The area under the receiver operating characteristic (ROC) curve (AUC). Values by different methods on the six datasets.

| Dataset | GRX | LRX | CRD | LSMAD | ERCRD | RCRDMF |
|---|---|---|---|---|---|---|
| AVIRIS-I | 0.9111 | 0,8194 | 0.9742 | 0.9717 | 0.9787 | **0.9911** |
| AVIRIS-II | 0.9403 | 0.8276 | 0.9357 | 0.9724 | 0.9798 | **0.9861** |
| AVIRIS-III | 0.8710 | 0.8326 | **0.9685** | 0.9308 | 0.9165 | 0.9616 |
| Cri | 0.9134 | 0.6779 | 0.7220 | 0.9236 | 0.9141 | **0.9943** |
| ABU-airport-2 | 0.8404 | 0.9492 | 0.9443 | 0.9438 | 0.9220 | **0.9759** |
| Salinas | 0.8872 | 0.9499 | 0.9075 | 0.9481 | 0.9422 | **0.9607** |

**Table 2.** The detection time by different methods on the six datasets.

| Dataset | GRX | LRX | CRD | LSMAD | ERCRD | RCRDMF |
|---|---|---|---|---|---|---|
| AVIRIS-I | 0.0914 | 60.7072 | 90.6277 | 13.0894 | 0.6448 | 3.5191 |
| AVIRIS-II | 0.0608 | 56.3844 | 60.2825 | 8.2589 | 0.4245 | 0.7417 |
| AVIRIS-III | 0.2786 | 219.3801 | 338.3894 | 47.6509 | 2.0212 | 13.8526 |
| Cri | 0.1558 | 101.2296 | 387.3111 | 35.6779 | 2.2044 | 178.7775 |
| ABU-airport-2 | 0.0769 | 56.4996 | 62.4306 | 11.7918 | 0.6671 | 0.9319 |
| Salinas | 0.2304 | 169.9561 | 191.2008 | 39.1329 | 2.2376 | 36.6195 |

For the AVIRIS-I dataset, the color detection result maps of different approaches are presented below in Figure 5. From this figure, we can determine that CRD, LSMAD, ERCRD, and RCRDMF methods can detect the locations and shapes of the three airplanes. The results obtained by the RCRDMF method are seen to be the clearest. Moreover, the corresponding ROC curve and background-anomaly separation map are presented in Figure 6. We can see that the ROC curve of RCRDMF method is far closer to the top left; in addition, the AUC value obtained by RCRDMF reaches 0.9911, much higher than the others. The gap between background and anomaly is large. The time cost for RCRDMF is 3.5191, which is faster than the CRD method. These results illustrate the superiority of the proposed RCRDMF method.

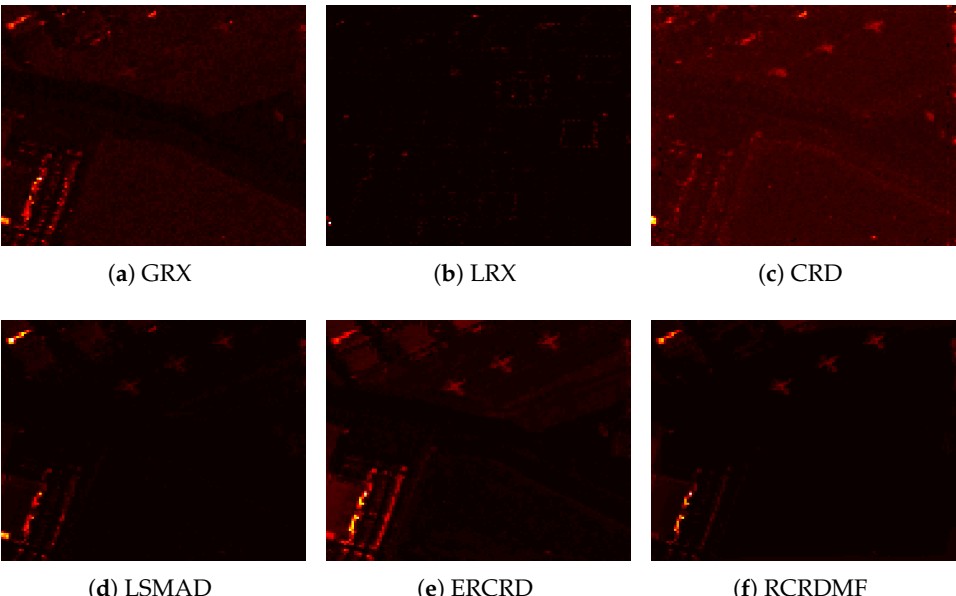

**Figure 5.** Color detection maps obtained by different algorithms for the AVIRIS-I dataset. (**a**) global RX (GRX), (**b**) local RX (LRX), (**c**) CRD, (**d**) LRaSMD-based Mahalanobis distance method (LSMAD), (**e**) ERCRD, (**f**) RCRDMF.

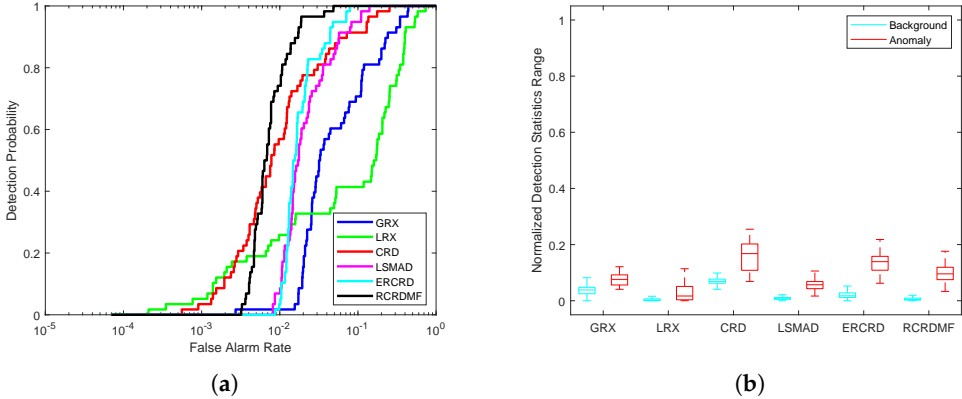

**Figure 6.** Detection accuracy evaluation for the AVIRIS-I dataset. (**a**) Receiver operating characteristic (ROC) curves. (**b**) Normalized background-anomaly separation maps.

For the AVIRIS-II dataset, the color detection result maps of different approaches are presented in Figure 7. The GRX, LSMAD, ERCRD, and RCRDMF methods can clearly detect the locations and shapes of the three airplanes. Among them, GRX, LSMAD, and ERCRD mistakenly identify many background points as anomalies, while RCRDMF can obtain a clearer detection result. Figure 8, Table 1, and Table 2 present the corresponding quantitative result. We can make the following observations. Firstly, the ROC curve of this method is much closer to the top left. The corresponding AUC value of RCRDMF is 0.9861, much higher than the others. Secondly, RCRDMF obtains an obvious background-anomaly separation map. Finally, RCRDMF requires 0.7417 s to complete detection, while the CRD method needs 60.2825 s. Therefore, RCRDMF performs best in terms of both qualitative and quantitative results.

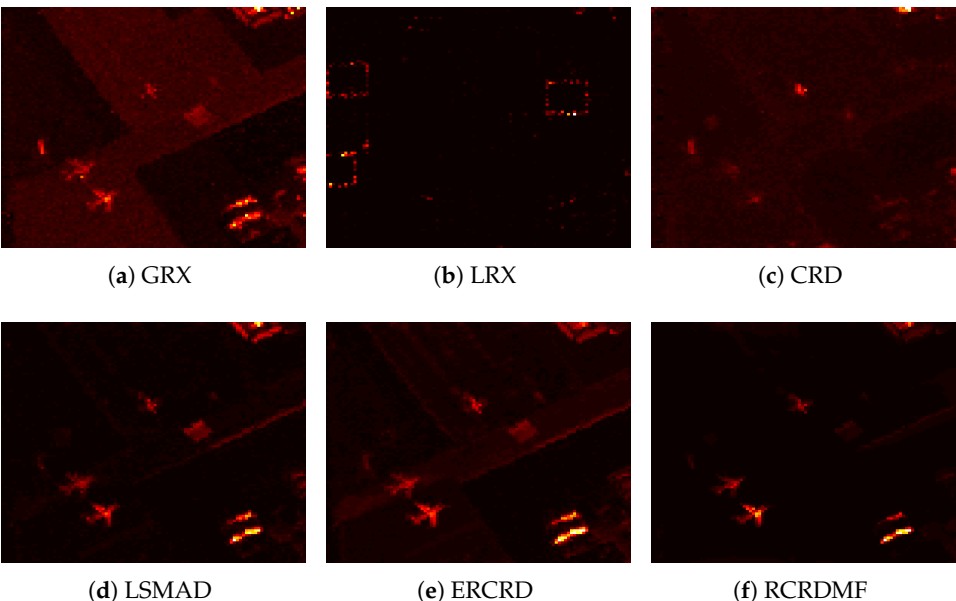

**Figure 7.** Color detection maps obtained by different algorithms for the AVIRIS-II dataset. (**a**) GRX, (**b**) LRX, (**c**) CRD, (**d**) LSMAD, (**e**) ERCRD, (**f**) RCRDMF.

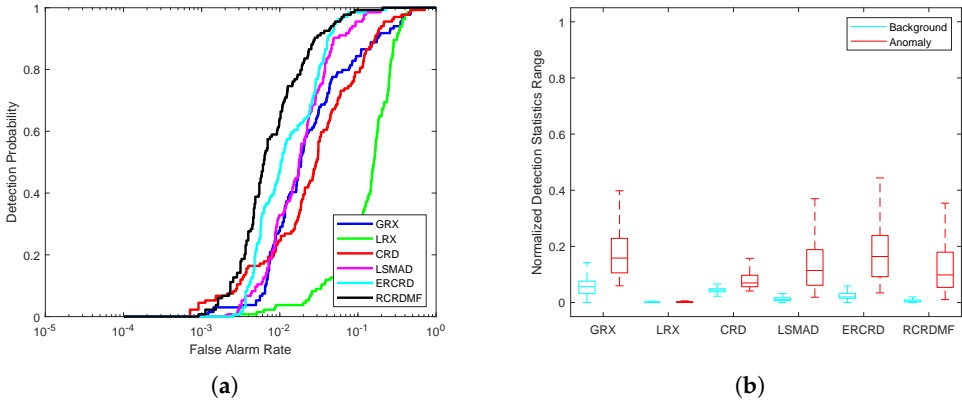

**Figure 8.** Detection accuracy evaluation for the AVIRIS-II dataset. (**a**) ROC curves. (**b**) Normalized background-anomaly separation maps.

For the AVIRIS-III dataset, the color detection result maps of different approaches are presented in Figure 9. The CRD, ERCRD, and RCRDMF methods can detect the locations and shapes of the three airplanes, while the others are unable to do so. Figure 10, Table 1, and Table 2 present the corresponding quantitative results. We can determine that the AUC values obtained by the CRD and RCRDMF methods are very close. However, RCRDMF requires only 13.8526 s to complete detection, while CRD needs 338.3894 s. Therefore, RCRDMF can obtain a reasonable detection result with little time.

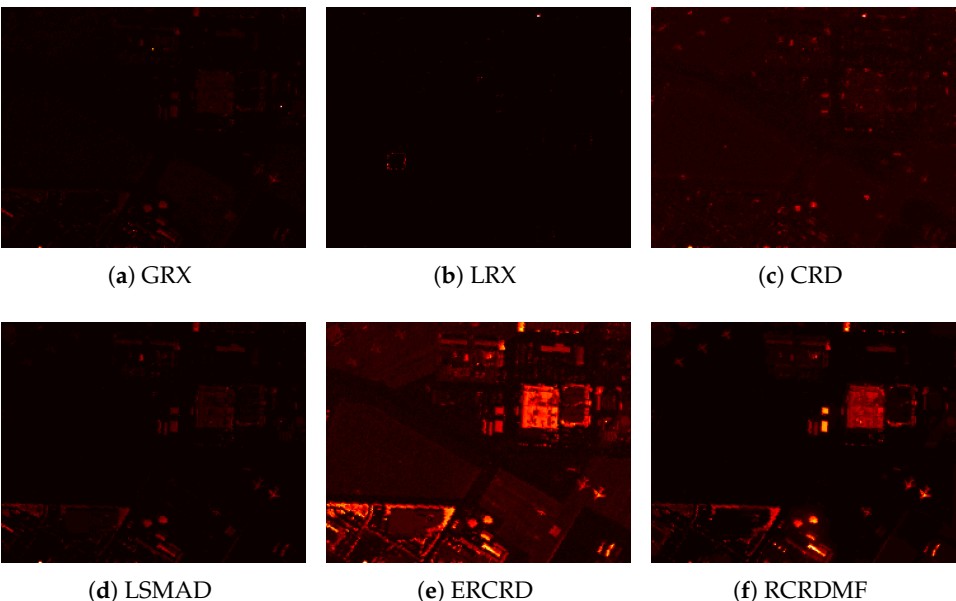

(**a**) GRX        (**b**) LRX        (**c**) CRD

(**d**) LSMAD        (**e**) ERCRD        (**f**) RCRDMF

**Figure 9.** Color detection maps obtained by different algorithms for the AVIRIS-III dataset. (**a**) GRX, (**b**) LRX, (**c**) CRD, (**d**) LSMAD, (**e**) ERCRD, (**f**) RCRDMF.

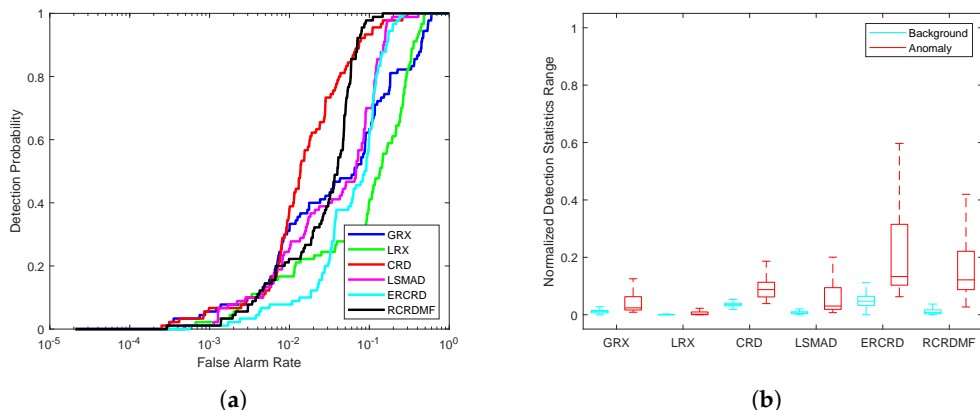

(**a**)                           (**b**)

**Figure 10.** Detection accuracy evaluation for the AVIRIS-III dataset. (**a**) ROC curves. (**b**) Normalized background-anomaly separation maps.

For the Cri dataset, the color detection result maps of different approaches are presented in Figure 11. Only the LRX method is unable to find the anomaly. LSMAD, ERCRD, and RCRDMF can clearly detect the locations and shapes of the ten rocks, while the result obtained by the CRD method is fuzzy. The corresponding quantitative results are presented in Figure 12, Tables 1 and 2. We can see that the ROC curve and AUC value obtained by the RCRDMF method are much better than those of the other methods. The AUC value of the RCRDMF method can reach 0.9943, which is 0.0707 higher than the second method LSMAD. Moreover, the background-anomaly separation map of RCRDMF is also very clear.

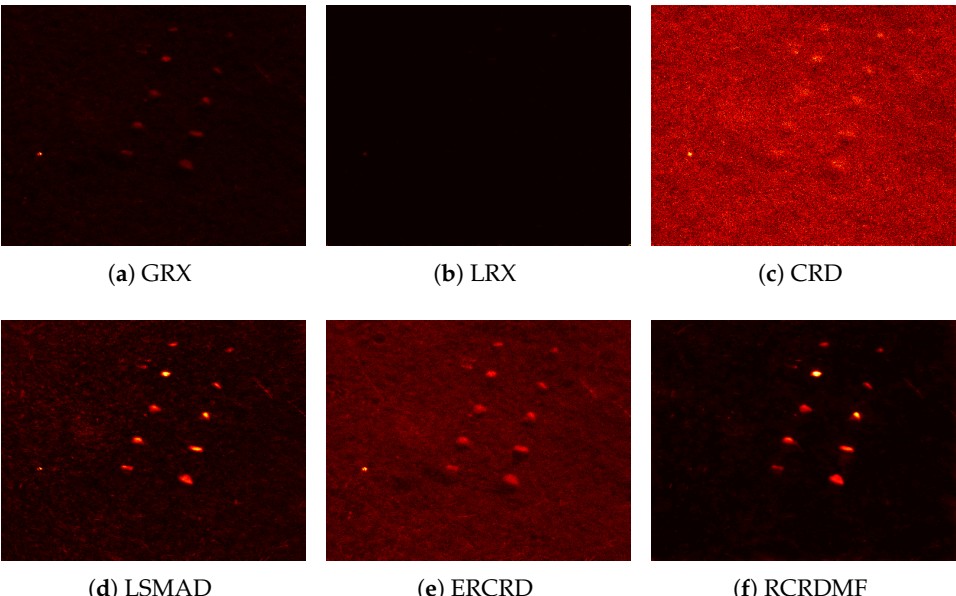

(**a**) GRX (**b**) LRX (**c**) CRD

(**d**) LSMAD (**e**) ERCRD (**f**) RCRDMF

**Figure 11.** Color detection maps obtained by different algorithms for the Cri dataset. (**a**) GRX, (**b**) LRX, (**c**) CRD, (**d**) LSMAD, (**e**) ERCRD, (**f**) RCRDMF.

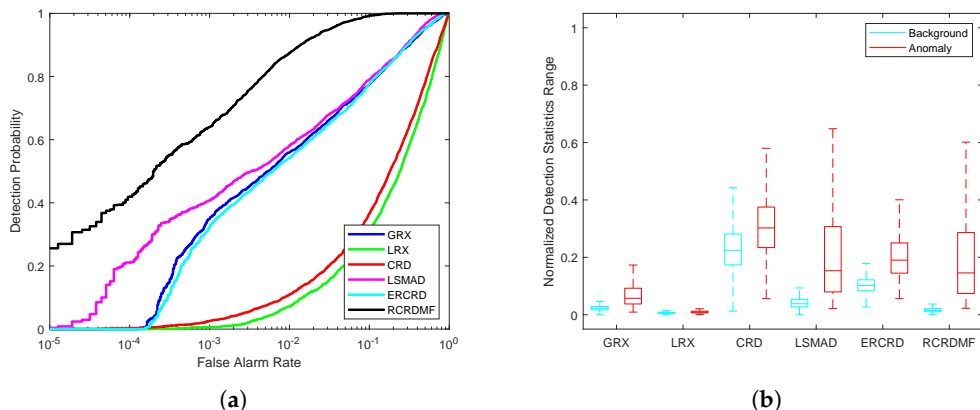

(**a**) (**b**)

**Figure 12.** Detection accuracy evaluation for the Cri dataset. (**a**) ROC curves. (**b**) Normalized background-anomaly separation maps.

For the ABU-airport-2 dataset, the color detection result maps of different approaches are presented in Figure 13. Although all six methods can detect the locations and shapes of the two airplanes, the corresponding quantitative results of RCRDMF are superior overall, as can be seen in Figure 14. The ROC curve of the RCRDMF method is far closer to the top left. In addition, it can be seen from Table 1 that the corresponding AUC value of the RCRDMF method is 0.9759, which is 0.0316 and 0.0539 higher than CRD and ERCRD method, respectively. The background-anomaly separation map of RCRDMF method is also very clear (see Figure 14b). The time cost of different methods are presented in Table 2. RCRDMF needs 0.9319 s to complete detection, while the CRD method needs 62.4306 s. Therefore, RCRDMF performs best in terms of the qualitative and quantitative results.

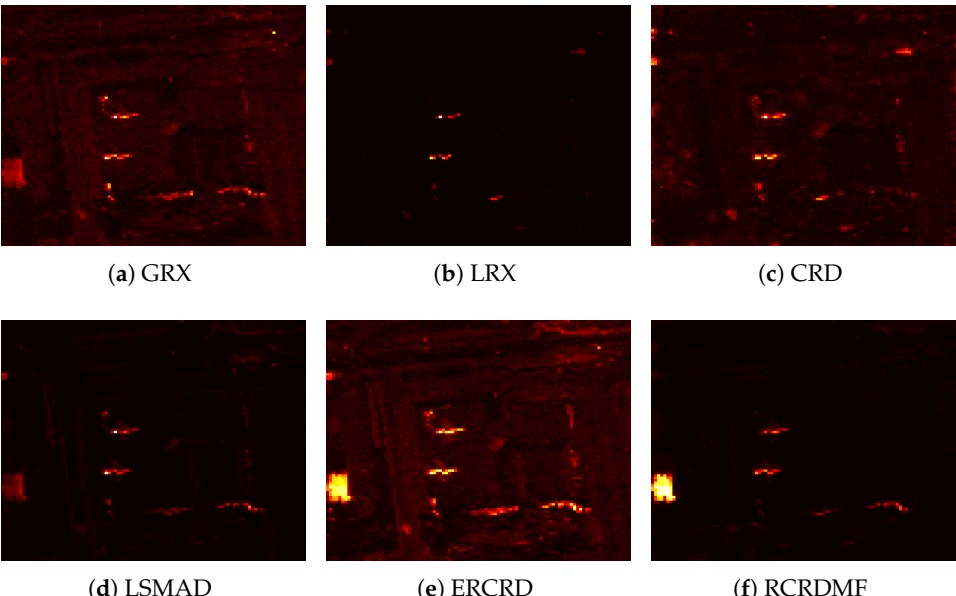

(**a**) GRX       (**b**) LRX       (**c**) CRD

(**d**) LSMAD       (**e**) ERCRD       (**f**) RCRDMF

**Figure 13.** Color detection maps obtained by different algorithms for the abu-airport-2 dataset. (**a**) GRX, (**b**) LRX, (**c**) CRD, (**d**) LSMAD, (**e**) ERCRD, (**f**) RCRDMF.

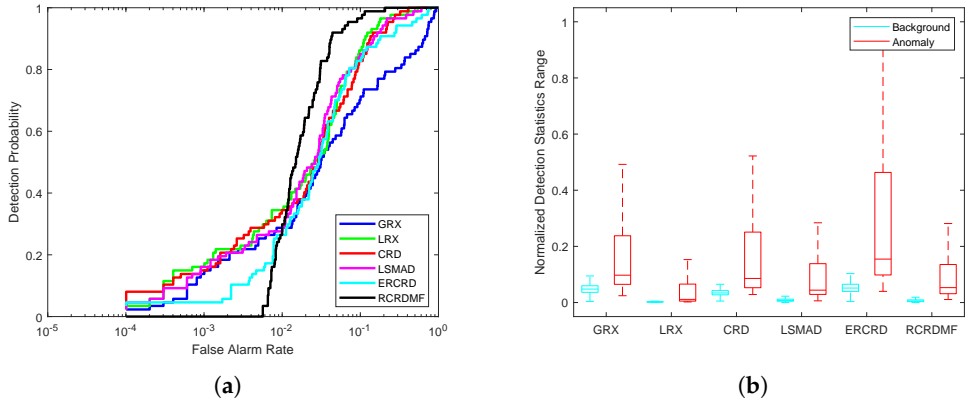

(**a**)                              (**b**)

**Figure 14.** Detection accuracy evaluation for the abu-airport-2 dataset. (**a**) ROC curves. (**b**) Normalized background-anomaly separation maps.

The color detection result maps of different approaches on the Salinas dataset are presented in Figure 15. The corresponding ROC curve is shown in Figure 16a. As the figures show, the ROC curve of the proposed RCRDMF method is much closer to the top left. Table 1 shows that the AUC value achieved by RCRDMF is 0.9607, which is much higher than that of the others. Figure 16b presents the background-anomaly separation maps of different methods; among them, RCRDMF can achieve a larger gap than the others. Table 2 further shows the time cost of each method. RCRDMF needs 36.6195 s to complete detection, while the CRD method needs 191.2008 s. Therefore, RCRDMF achieves a better detection result.

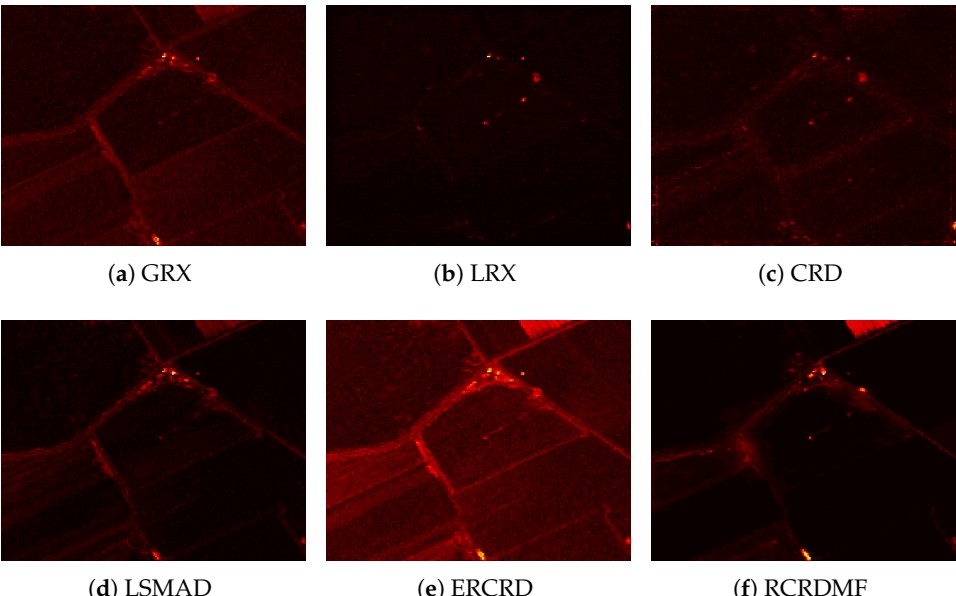

(**a**) GRX        (**b**) LRX        (**c**) CRD

(**d**) LSMAD        (**e**) ERCRD        (**f**) RCRDMF

**Figure 15.** Color detection maps obtained by different algorithms for the Salinas dataset. (**a**) GRX, (**b**) LRX, (**c**) CRD, (**d**) LSMAD, (**e**) ERCRD, (**f**) RCRDMF.

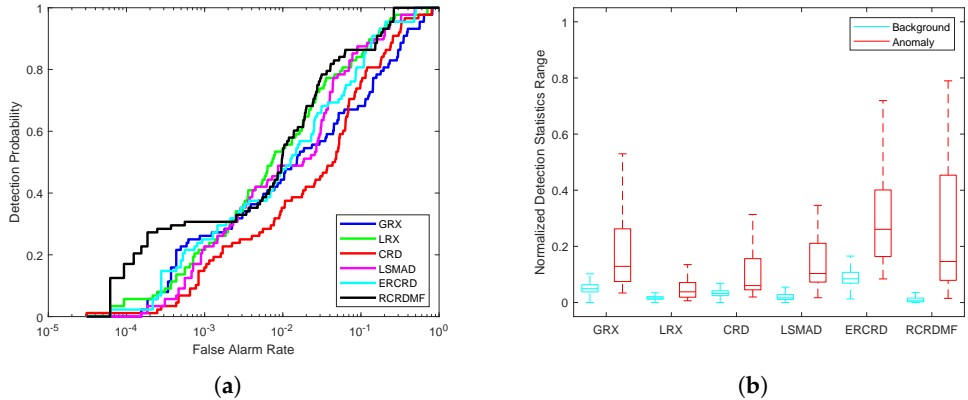

(**a**)                             (**b**)

**Figure 16.** Detection accuracy evaluation for the Salinas dataset. (**a**) ROC curves. (**b**) Normalized background-anomaly separation maps.

## *3.2. Parameter Analysis*

In this subsection, we conduct experiments to determine the influence of parameters referred to in the ERCRD and RCRDMF methods. We select a small dataset (AVIRIS-I) and a large dataset (Cri) for these specific experiments. In the ERCRD and RCRDMF methods, there are two major parameters: the number of random samples $r$ and the ensemble size $T$. In addition, the quantitative index (AUC value) is adopted to assess the performance of the ERCRD and RCRDMF methods.

On the AVIRIS-I dataset, the relationships between the number of random samples $r$ and the AUC value are plotted in Figure 17a. Compared with the ERCRD model, the proposed RCRDMF method is much more robust. The AUC value obtained by the RCRDMF method is also higher than that of the ERCRD method for different number of random samples $r$. The relationship curves between the ensemble size $T$ and the AUC value are further plotted in Figure 17b. RCRDMF method always outperforms the ERCRD method. As the ensemble size $T$ increases, the detection results become more stable.

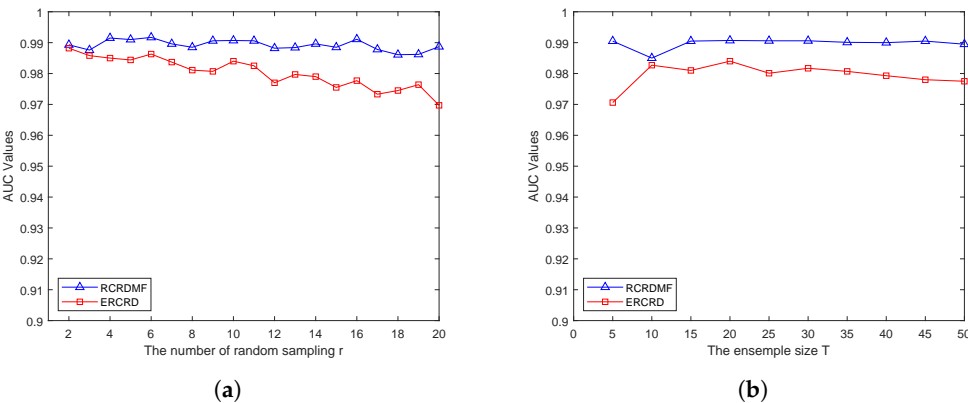

**Figure 17.** Parameter effect of (**a**)the number of random sampling *r* and (**b**) the ensemble size *T* on the AVIRIS-I dataset.

On the Cri dataset, the parameter effect of the number of random samples *r* and the ensemble size *T* are presented in Figure 18. From Figure 18a, we can see that RCRDMF still performs better than ERCRD in different situations. Figure 18b also presents the superiority of the RCRDMF method.

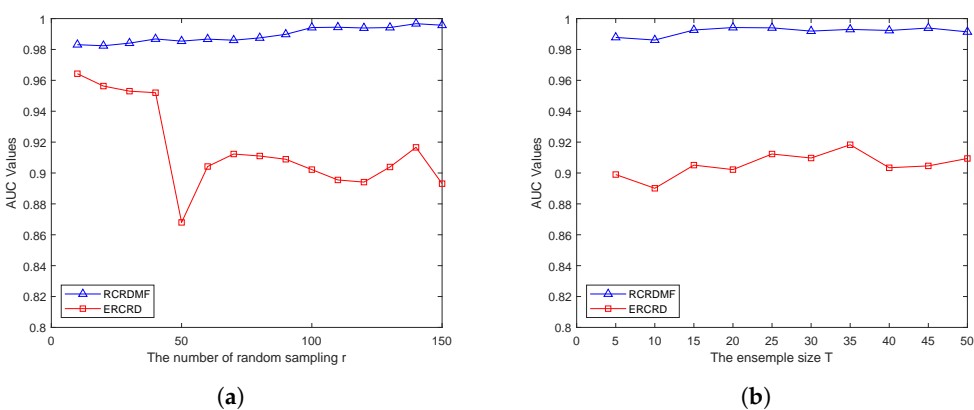

**Figure 18.** Parameter effect of (**a**) the number of random sampling *r* and (**b**) the ensemble size *T* on the Cri dataset.

### 3.3. Feature Weight

The proposed RCRDMF use four types of features for anomaly detection. There will be one group of weight for these features generated after each run of Algorithm 1, line 2~8. Since the *T* was set as 20 on each dataset in our experiment, Table 3 lists ten of the twenty groups of weight on AVIRIS-I and Cri datasets, respectively. From the Table 3, we can see that the weights are similar on a certain dataset and are quite different between the two datasets.

**Table 3.** Ten groups of weight for the four features on AVIRIS-I and Cri datasets.

| AVIRIS-I | | | | Cri | | | |
|---|---|---|---|---|---|---|---|
| **Spectral** | **Gabor** | **EMP** | **EMAP** | **Spectral** | **Gabor** | **EMP** | **EMAP** |
| 0.1522 | 0.4298 | 0.1404 | 0.2776 | 0.2499 | 0.0001 | 0.2167 | 0.5333 |
| 0.1539 | 0.4495 | 0.1394 | 0.2572 | 0.2538 | 0.0001 | 0.2019 | 0.5443 |
| 0.1514 | 0.4508 | 0.1465 | 0.2513 | 0.2438 | 0.0001 | 0.2080 | 0.5481 |
| 0.1461 | 0.4835 | 0.1265 | 0.2439 | 0.2493 | 0.0002 | 0.2092 | 0.5414 |
| 0.1415 | 0.4584 | 0.1404 | 0.2596 | 0.2389 | 0.0001 | 0.2062 | 0.5548 |
| 0.1878 | 0.3575 | 0.1630 | 0.2916 | 0.2546 | 0.0001 | 0.2233 | 0.5221 |
| 0.1418 | 0.4335 | 0.1511 | 0.2736 | 0.2411 | 0.0001 | 0.2105 | 0.5484 |
| 0.1708 | 0.4344 | 0.1358 | 0.2591 | 0.2557 | 0.0001 | 0.2152 | 0.5290 |
| 0.1728 | 0.4466 | 0.1394 | 0.2413 | 0.2595 | 0.0001 | 0.2162 | 0.5242 |
| 0.1593 | 0.4072 | 0.1515 | 0.2821 | 0.2519 | 0.0001 | 0.2247 | 0.5233 |

## 4. Discussion

In this paper, we propose a novel anomaly detection approach named Random Collective Representation-Based Detector with Multiple Feature (RCRDMF). The main motivation of our approach is to make full use of the information provided by HSI images in the field of anomaly detection, which is different with the classic methods only use the spectral features, e.g., RX, CRD, etc. Fortunately, there are several widely used features can be applied for HSI images, such as Gabor, EMP, and EMAP features. We extract these features and find the approach to utilize the correlative information for anomaly detection. In spite of the different generation methods and different dimensions, all the features can be corresponded to the pixels of the original image one by one. Therefore, it is more appropriate to achieve our purpose based on a representation-based method. As it is known that CRD operates on pixels, thus, it can benefit a lot with our approach. On the other hand, CRD consumes a lot of time; thus, the use of multiple features will lead to unacceptable complexity. To solve this problem, we refer to another method named ERCRD. The ERCRD uses random background points to replace the sliding dual window model, which greatly improves the running time. As can be seen from Table 2, the running time of the ERCRD is particularly fast. In our method, the same strategy of the ERCRD is utilized on each selected feature and these results are combined together in the final expression. The experiment results show that our RCRDMF method can obtain better AUC than the others in most situation. These results verifies that the use of multiple features does improve the anomaly detection accuracy because of the more affluent information. One exception is that the AUC of our method is not better but close to the CRD on the AVIRIS-III dataset, but our method has the obvious advantage on running time. In fact, from Figure 4c,f, it can be found that some backgrounds in the image are more special than the anomalies. It can be seen from Figure 9f that our method can clearly detect the anomalies. The reason for the unsatisfactory result is that the special backgrounds were also detected as the anomalies. Since these special backgrounds are mostly fixed buildings, the most possible solution is to remove these fixed pixels in practical application.

According to the strategy of selecting the background points randomly in our algorithm, the number of the random selected points $r$ should be chosen carefully. Similar to ERCRD, we assume that the pixels in the background can be approximately represented by the randomly selected pixels in the HSI image, while this is not the case for anomalies. As the number $r$ should be depended on the size of the dataset, we tried to use a fixed ratio of data size to select the $r$ in the design at the begining. It seems that no need to adjust the number is a great strategy for our algorithm. However, when the dataset is large enough, the $r$ becomes larger at the same time since the number of pixels is quadratic growth, which consumes much time. So, it is improper to use a fixed ratio. In Section 3.2, we use experiments to illustrate the effect of different $r$ on the AUC. It can be seen from Figures 17a and 18a that on a small dataset, like AVIRIS-I, a small $r$ can achieve a good result, and the result does not change significantly with the change of $r$; and on a large

dataset, like Cri, the larger $r$, the better result. Nevertheless, $r = 100$ can achieve a good result for Cri dataset. Based on the above discussion, we empirically set $r = 10$ for small datasets (AVIRIS-I and AVIRIS-II); $r = 50$ for medium datasets (AVIRIS-III, Salinas); and $r = 100$ for the large dataset (Cri), which can achieve a significant accuracy with satisfied time consumption. Specially, for the small dataset ABU-airport-2, there is a worse result when $r = 10$. It mainly because the background of ABU-airport-2 is quite complex, as is seen from Figure 4h. In this situation, the background pixels also need more random points to be represented, which is in line with our initial assumption. In our experiment, the AUC of ABU-airport-2 becomes the highest among the all methods when $r = 100$.

Because of the randomness of the algorithm, it is definitely unstable to choose random background points only once for detection. Inspired by the ensemble method of the ERCRD, we repeat the process of randomly selecting points and calculating the anomaly score for $T$ times and then ensemble the results. This strategy can eliminate the random effects and make the algorithm more stable. In Section 3.2, we use experiments to illustrate the effect of $T$ on the AUC. It can be seen from Figures 17b and 18b that, with $T$ increases, the result becomes more stable. We set $T = 20$ in practice.

In this paper, we use the machine learning method to learn a suitable weight for multiple features adaptively. This model achieve the effect that the more important feature weight the larger automatically. As can be seen from the Table 3, the weights are similar on a certain dataset and are quite different between different datasets. This illustrates that the importance of one feature is similar on the same data and is distinguishing between different datasets, as well as further illustrates that our method makes full use of the information of these features.

In related literature, the performance of an anomaly detection approach is usually evaluated by the ROC curve, AUC, and the separation between the anomalies and the background, which does the same as this article. It is easy to understand that a larger of the anomaly score, the more likely the corresponding pixel is an anomaly. However, these evaluation indicators do not tell us how to appropriately set a threshold to determine whether a pixel is an anomaly in practice. In fact, setting a appropriate threshold need to comprehensively consider the relationship between the requirements of false alarm rate and accuracy, and combine the effective prior knowledge. In more general situation, it is reasonable to apply a certain method, e.g., histogram method, to divide the scores into two patterns; thus, the threshold is the dividing line between the two parts.

## 5. Conclusions

In this paper, we propose a novel anomaly detection approach named Random Collective Representation-Based Detector with Multiple Feature (RCRDMF). The spectral feature, Gabor feature, EMP feature, and EMAP feature are utilized to construct the corresponding ERCRD detector. Next, the adaptive weight approach is adopted to calculate the weight of each feature. The advantages of RCRDMF are as follows: First, the combination of multiple spectral and spatial features can improve the detection accuracy. Secondly, the detection speed can be accelerated with the help of random background modeling. Finally, the adaptive weight approach is proposed to calculate the weight for each feature, which removes the need to tune the weight parameter. Experiments on six real hyperspectral images illustrate the superiority of our proposed RCRDMF in terms of detection accuracy and speed. In addition, the parameter sensitivity experiments also prove the robustness of the RCRDMF model.

**Author Contributions:** Methodology, Z.L. and F.W.; Project administration, F.W.; Software, Z.L.; Visualization, H.H.; Writing—original draft, F.H. and Z.L.; Writing—review & editing, Z.L., H.H. and W.Y. All authors have read and agreed to the published version of the manuscript.

**Funding:** This research received no external funding.

**Institutional Review Board Statement:** Not applicable.

**Informed Consent Statement:** Not applicable.

**Acknowledgments:** We thank the editors and reviewers for their insightful comments.

**Conflicts of Interest:** The authors declare no conflict of interest.

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
