# Peer review of "Random Collective Representation-Based Detector with Multiple Features for Hyperspectral Images"

_remotesensing, doi:10.3390/rs13040721_

Round 1

Reviewer 1 Report

I have found this paper clear and quite well written.
The contents of this manuscript provide interesting and original ideas to improve the resolution of the anomaly detection problem.
The Authors have provided a well-structured exposition of their material.
The content is described with an acceptable level of detail to understand the topic, techniques and results. But there is still room for improvement.
The analysis provided and corresponding results are fully appropriate to the text and its content.
The list of references to the literature related to the field is also quite appropriate.

Overall, the content is acceptable and the work carried out is original.
However, I have some comments to help the authors to improve further the quality of the manuscript content.

1) Please refine the following phrase "The primary representation-based methods have:..."

2) I think it necessary in the Introduction to better introduce and discuss more in depth the meaning of the weight for each feature.

3) Recall more in detail but still briefly how the extended attribute profiles (EAPs) are obtained from the morphological attribute filters.

4) Refine ", in each of that has 100 × 100 pixels", "All of these can illustrate"

5) I think it would be interesting to add for each experiment the final value of the alpha_v parameter at the output of RCRDMF algorithm 1 and to comment on the trends observed (the initialization being uniform at the input)

6) How to appropriately set the decision threshold in practice remains to be described especially for the proposed approach, once (16) is obtained. This is currently missing from the manuscript.

7) Could this setting call into question the hierarchy between methods, established on the basis of the experimental results given here?

There are a number of typos to correct (check carefully please):
an improtant role
et.al
tomodel
CRBORAD method estimate ...
RCRDMF model also cost ...
to result the extend attribute profils
AVIRIS-III: The same as AVIRIS-I and AVIRIS-III datasets
dataset[kang]
can detect the the locations
The corresponding quantitative result are described
The the spectral feature,

Reviewer 2 Report

The paper introduces a method to detect anomalies in hyperspectral images. It is based on a pixel-based approch which combines several derivatives and compares them pixel by pixel in the pixels envornment. The approach is not very innovative but the authors could demonstrate its effectiveness.

English should be re-checked by a native speaker.

Section 2 should either be re-named or re-organized since it does not really show related work. At least I am missing substantial references here.

Terms like:"... hot research area ..." must not be used.

Reviewer 3 Report

This paper proposes an method, called random collective representation-based detector with multiple feature (RCRDMF), for hyperspectral anomaly detection. Roughly speaking, the RCRDMF is an improved version of the existing collaborative representation-based detectors. It considers both spectral and spatial information (such as Gabort featire, EMP, and so on), and a random sampling strategy of background pixels, to achieve more efficient and accurate detection performance. The experiments conducted on 6 real hyperspectral data show the promise of the effectiveness of RCRDMF.

The motivation and background are well-introduced, particularly the progress of the development of modern anomaly detection algorithms. The methodology part and theoretical part sound reasonable. A comprehensive comparison in detection rate and computing time, as well as the extended parameter analysis is included in the experiment section. The reviewer thinks this paper provides a certain level of contribution, is worthy to be published. However, there are few questions/issues encouraged to be fixed. The comments are as follows:

  1. In RCRDMF, r presents the number of background pixels to be selected. In experiment, the authors set r for each hyperspectral data empirically (in Line 282-283). How did the authors decide the value of r? This is very important for the practicality of the algorithm.

  1. Once r is determined, the RCRDMF “randomly”selects r background pixels from the entire image. Because of the random selection, the result would be different at each run. How did the authors report the AUC results? Are the AUC values shown in Figs 5,7,9,11,13,15 the averaged values of multiple runs? Also, did the authors analyze the performance variation for each image? In a normal condition, the AUC value should be reported as [Mean value +- Standard deviation].

  1. Please explain how to draw “Normalized baclkground-anomaly separation maps” shown in Figs 5,7,9,11,13,15.

4. It is encouraged to add a diagram to introduce RCRDMF in the beginning of Section 3.

Reviewer 4 Report

In this paper a novel anomaly detector fro hyperspectral imaging is proposed. Ths paper includes a complete state of the art review together with a clear algorithm description and a comparison with several methods. The main issue with the paper is the level of the use of the English language is unsatisfactory to be included in this publication. Authors whose primary language is not English are advised to seek help in the preparation of the paper. Please also consider the following aspects to be amended:

  • Line 170: It is mentioned that if a threshold is exceeded, xi is regarded as an anomalous pixel. However it is not mentioned how this threshold is defined. Please clarify.
  • Lines 280-283: Several values for the number of randomw sampling "r" are proposed for state of the art methods. Please clarify why these valeus are selected.
  • AUC Values in section (b) of Figures 5, 7, 9, 11, 13 and 15 could be included ina single Table similar to Table 1 for detection time. Therefore, sections (a) and (c) of these figure may be enlarged for clearness.
  • Conclusions should be extended to highlight the main results of the paper.  

Round 2

Reviewer 1 Report

The authors have corrected and revised the content of their manuscript appropriately in relation to the remarks and suggestions raised in the previous round.
The quality of the manuscript has improved and the manuscript is now in a form acceptable for publication.